# Protocol for designing and testing the effectiveness of a post caesarean section home care guide in preventing surgical site infection in Central Tanzania

Mwajuma Bakari Mdoe[1]*, Lilian Teddy Mselle[2], Stephen Mathew Kibusi[3]

1 Department of Clinical Nursing, School of Nursing and Public Health, University of Dodoma, Dodoma, Tanzania, 2 Department of Clinical Nursing, School of Nursing, Muhimbili University of Health and Allied Sciences, Dar es Salaam, Tanzania, 3 Department of Public Health, School of Nursing and Public Health, University of Dodoma, Dodoma, Tanzania

* mwajuma_mdoe@yahoo.com

**Data Availability Statement:** The data from this study are owned by the Muhimbili University of Health and Allied Science- Research Ethics

## Abstract

### Background

The advantages of caesarean section (CS) in managing obstetric emergencies are abundant, but it is associated with several complications including surgical site infection (SSI). SSI significantly contributes to maternal morbidity and mortality rates. Mothers often do not receive adequate information about their at-home post-delivery care. Also, guidelines on post-CS care worldwide typically do not include home care recommendations. Because of the increased rate of CS and space constraints in the hospitals, mothers are often discharged home within 48 hours after CS. Therefore, it is anticipated that using an evidence-based home care guide would provide instruction to the mothers and is likely to prevent postpartum complications and promote the well-being of both the mother and the newborn.

### Aim

To design and test the effectiveness of a post-CS home care guide in preventing SSI in central Tanzania.

### Methodology

This is a sequential exploratory mixed-method interventional study conducted in two regional referral hospitals in central Tanzania. A qualitative study will be conducted to explore the experiences of nurse midwives, mothers who had caesarean deliveries and their caretakers regarding the care of mothers and newborns at home. The findings will inform the development of a post-CS home care guide. Following a series of validations of the guide, research assistants will employ the guidelines to instruct post-CS mothers about home care as part of the intervention. Thirty participants will purposively be recruited for the qualitative study and a random sample of 248 nurse-midwives and 414 post-CS mothers to assess the effectiveness of the guide in improving knowledge of home care and preventing

Committee (MUHAS-REC). However, they may be requested from the Directorate of Research and Publication. Email: drp@muhas.ac.tz.

**Funding:** The source of fund or this study is the University of Dodoma, that covered both the tuition fee for the PhD and the data collection costs. The funders had no role in study design, data collection and analysis, decision to publish, or preparation of the manuscript.

**Competing interests:** The authors have declared that no competing interests exist.

SSI. SPSS version 25 will be used to analyse quantitative data and content analysis, and ATLAS.ti will guide in analysing the qualitative data.

## Conclusion

The post-CS home care guide will provide instructions to post-CS mothers and their caretakers about the care of the mothers after CS to enhance their recovery.

## Introduction

Caesarean section (CS) is a common surgical intervention during foetal delivery, especially when vaginal delivery is associated with high risk to both the woman and the foetus. Globally, the rate of CS is on the rise as is evident from the doubled caesarean deliveries from 2000 to 2015. For example, the incidence of CS in the Middle East and North Africa increased from 19% in 2000 to 29.6% in 2015 [1]. The Demographic and Health Survey report of 2015 reported the prevalence of CS in Tanzania's mainland to be 6% [2]; however, anecdotal evidence shows that CS rates in its regional hospitals are as high as 48%. This is anticipated because regional hospitals usually receive referrals of mothers with obstetric complications that require CS.

Although CS is a critical intervention for managing complex obstetric issues, it may result in various complications including rupture of the uterus in subsequent pregnancies, injury of the nearby organs, postpartum anaemia and SSIs [3–5]. SSI is the most common complication associated with CS and contributes significantly to maternal morbidity and mortality [6]. Mothers who underwent CS have a 5 to 20 times increased risk of acquiring infections compared to mothers with spontaneous vaginal delivery [7]. In Tanzania, sepsis is reported to be the third direct cause of maternal death after haemorrhage and eclampsia [8]. SSIs from CS may lead to prolonged hospitalisation, increased healthcare costs, repeat operations to treat the infected wound, and endometritis, and may render the mother unable to care for her newborn and, in severe cases, cause maternal death [9–11]. With a global increase in CS cases there has been a proportional increase in the total number of SSIs [1, 3, 12, 13]. In Europe and the United States of America (U.S.A), SSI is the second most frequent complication of the majority of operations [14, 15]. In 2017, WHO reported that up to 20% of mothers who deliver by CS in Africa, develop SSIs [16]. In Tanzania, the prevalence of SSI attributable to CS ranges from 11% to 48% [10, 17–19].

SSI may occur as a result of repeated vaginal assessments, contamination of the surgical wound due to improper care, longer operation duration, pre-existing diabetes, severe anaemia, and a lack of awareness of home care practices after hospital discharge [10, 17]. The guidelines issued by WHO for the prevention of SSI are as per the pre-operative, intraoperative and post-operative periods. The pre-operative recommendations include regular bathing, hair removal on the surgical site, mechanical bowel preparation and prophylactic oral antibiotics [16]. Intraoperative recommendations include perioperative blood glucose control, incision wound prolongation, appropriate timing for wound draining removal and advanced dressings are recommended as post-operative care [16]. In 2022, WHO updated its recommendations on maternal and newborn care during the post-natal period, details for home care recommendations especially for post-CS mothers, is lacking [20].

The Enhanced Recovery After Surgery (ERAS) Society's guidelines for postoperative care after CS recommends sham feeding, prevention of nausea and vomiting, provision of analgesics, nutritional care, glucose control, thromboembolism prophylaxis, early mobilisation, urinary drainage and discharge counselling on home care [21]. The guidelines emphasise

discharge counselling on home care for post-CS mothers. However, it does not provide clear and detailed instructions on the same. The authors of these guidelines also admit a paucity of studies addressing optimal counselling recommendations for post-CS care [21]. In Tanzania, there are no guidelines available for post-CS home care and published studies reporting care of mothers at home after caesarean section deliveries are lacking. Guidelines to instruct care at home are critical because most post-caesarean section mothers are typically discharged home within 48 hours [16, 22] to avoid overcrowding of the hospitals. Further home visits by a nurse midwife for monitoring the progress of the post-CS woman after discharge is uncommon. Therefore, family members at home are primarily involved in the care of mothers and babies. This is contrary to developed countries, where home visits and surveillance for post-CS mothers are routinely conducted [13]. Studies report a complete lack of or less information regarding home care after CS compared to home care after vaginaly delivery [23, 24]. The limited available information about home care after CS highly varies among different sources across the region [25]. Post-discharge instruction, especially for post-CS mothers has been a neglected aspect of post-CS care for a long time [21]. Even the existing advanced guidelines worldwide are severely lacking [16, 21].

The government of Tanzania has upgraded and established more than 400 health centres in the country to provide Comprehensive Emergency Obstetric and Newborn Care (CEmONC) services, thereby increasing access to CEmONC services. With increased access to CEmONC services and early discharge of post-CS mothers, it is estimated that the need for post-CS care will increase. Therefore, the availability of a standard post-CS home care guide to provide guidance to post-CS mothers is of paramount importance. The use of a post-CS home care guide is expected to improve the knowledge of post-CS mothers and nurse midwives on home care after CS, and ultimately reduce post-CS complications including SSI in the country.

## Conceptual framework

The self-care deficit theory [26] and Consolidated Framework for implementing Research (CFIR) [27] will guide the study. The self-care deficit model emphasises offering assistance to clients who can not care for themselves. The self-care can either involve total compensation, partial compensation, or education and support (total compensatory system, partially compensatory and educative-developmental system) [26, 28]. This study will focus on the education developmental system of the model that post-CS mothers should implement in order to manage their health through the instructions on care to be provided at home after CS.

The Consolidated Framework of Implementation Research (CFIR) consists of five domains; 1. Intervention 2. Inner environment 3. Outer environment 4. The individual involved in the intervention and 5. The implementation process of the intervention [27]. In this study, the intervention involves training nurse midwives and educating post-CS mothers. The inner environment is post-CS ward where education intervention will primarily take place and the outer environment is a post-CS mother's home environment, the place to implement the recommended home care practices. The implementation process will involve training nurse midwives first about the guide followed by the provision of education to post-CS mothers on home care by the trained nurses [27]. All the above-mentioned domains are needed for the effective implementation of the intervention as illustrated in the Fig 1.

## Aim and objectives

This study aims to design and test the effectiveness of a post-CS home care guide in increasing knowledge of home care and preventing postpartum complications following CS delivery in central Tanzania.

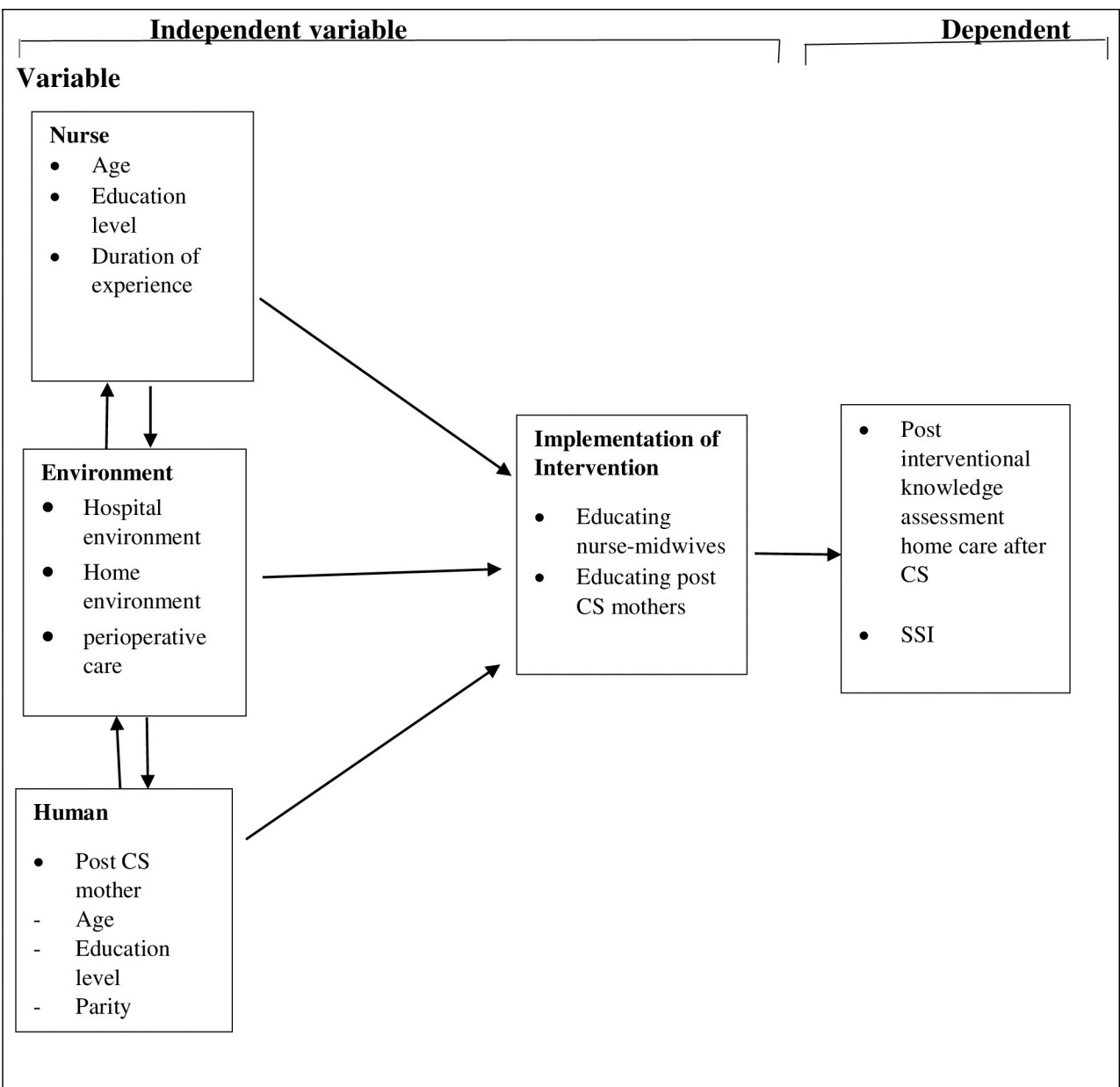

**Fig 1. Conceptual framework guiding the study, developed from self-care deficit theory and consolidated framework for implementing research.** The interaction of independent and depend varibles; Both the nurses, environment (both home and hospital environment), human that is post caesarean section mothers and the designed intervention are the independent variables that define the dependent variables that are knowledge on home cre acre Surgical Site Infection (SSI).

Objectives are to:

1. Develop and validate the post-CS home care guide

2. Assess the effectiveness of the post-CS home care guide in improving the knowledge of post-CS mothers and nurse midwives working at health facilities in central Tanzania.

3. Assess the effectiveness of a post-CS home care guide in preventing SSI in central Tanzania.

## Material and methods

### Study design

The study will employ a sequential exploratory mixed interventional study design, whereby qualitative data collection and analysis precedes the collection and analysis of quantitative data [29, 30]. The mixed method has been chosen because of its strength in portraying both qualitative and quantitative research and minimising the limitations of both approaches [30].

The study will have four phases; the first phase will involve conducting an integrative review of the home care recommendation for post-CS mothers. The second phase will explore: 1. Experiences of nurse-midwives on home care information provided to post-CS mothers 2. Experience of post-CS mothers on home care and information received after CS 3. Experience of caretakers on the care of post-CS mothers they provided at home after hospital discharge. The third phase will involve the development and validation of the post-CS home care guide by integrating results from phase one, phase two, and expert opinion, and pre-testing the guide. The fourth phase will involve intervention that is training nurse midwives on the guide and health education to post-CS mothers, and evaluation of the outcome of the intervention in increasing knowledge of home care and preventing SSI postpartum complications as shown in Fig 2.

### Study setting

This study will be conducted in the central part of Tanzania, specifically in Dodoma and Singida regions. Dodoma region has an area of 41,311 km$^2$ with a population of 2,083,588 and lies centrally in eastern central Tanzania. Singida region is within the central zone of Tanzania with 49,340 km$^2$ and a population of 1,370,637. The main economic activity in Singida is crop cultivation, especially sunflower, maize and groundnuts. Dodoma region has been chosen for implementing the intervention because of the reported high prevalence of SSI [19]. The Singida region has been chosen as a control region as it matches Dodoma geographically. That is,

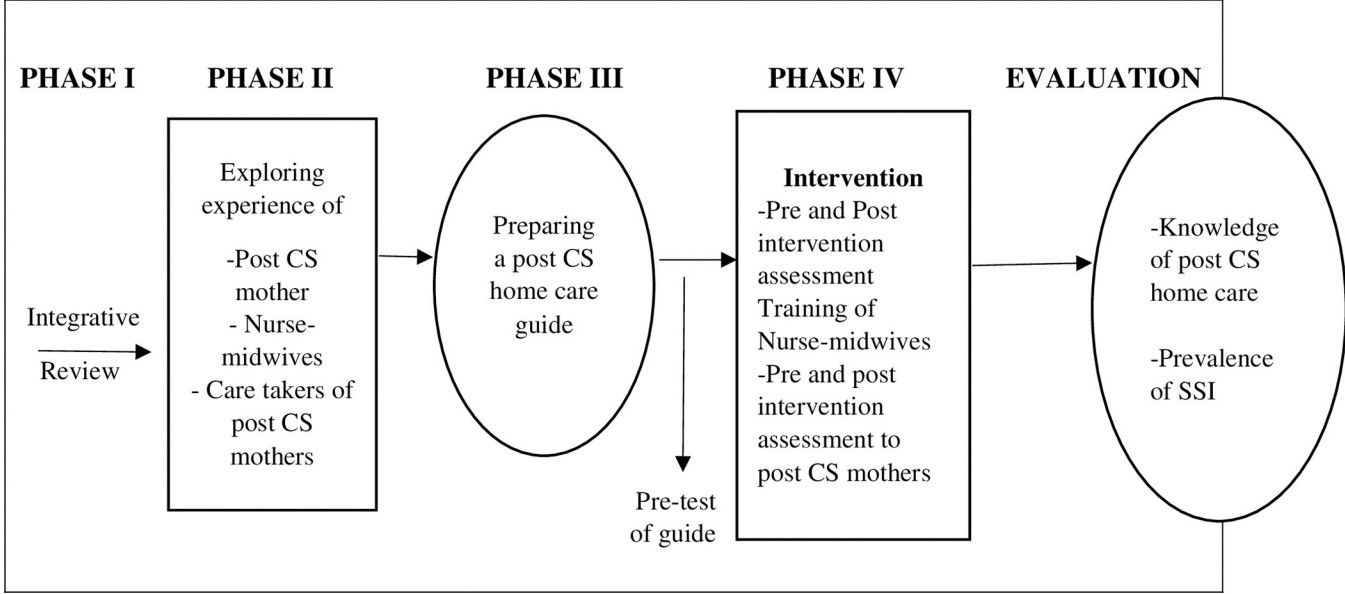

**Fig 2. The study phases.** Phase one of intergartive review and phase two of qualitative study will develop the guide in phase three; and will be tested in phase four on preventing SSI and knowledge after being pre-tested.

it is located in central Tanzania, and both of these regions' regional hospitals provide Comprehensive Emergency Obstetric and Newborn care (CEmONC).

## Phase I: Conducting integrative review

**Objective.** To evaluate and describe evidence-based content related to care provided to mothers at home following a caesarean section delivery.

**Methodology.** The integrative review is the comprehensive approach of reviews that allows for including experimental and non-experimental studies for an effective understanding of a phenomenon [31]. In this study, the integrative review will offer a synthesis of a wide variety of literature such as articles of cross-sectional studies, interventional, cohort studies; guidelines; protocols and case reports. The review will be guided by six stages as described by Souza and Carvalho [31], which include; 1. Preparing the guiding questions 2. Searching or sampling the literature 3. Data collection 4. Critical analysis of the studies included 5. Discussion of the results 6. Presentation of the integrative review [31].

**Guiding questions.** The questions that this review intends to answer include:

1. What are the home-based wound care practices for preventing SSI among post-CS mothers recommended in the literature?

2. What is the home-based nutritional recommendation in the literature for enhancing the recovery of post-CS mothers?

3. What are the recommended home-based hygienic practices for preventing SSI among post-CS mothers in the literature?

4. What are the home based recommended exercises existing in the literature to enhance the recovery of post-CS mothers?

5. What are the recommended environmental characteristics existing in the literature to enhance the recovery of post-CS mothers?

**Search words.** The keywords for searching literature will be guided by the questions mentioned above. These include the following MeSH terms:

1. "surgical site infection" OR "SSI" OR "wound sepsis" OR "wound infection" OR "wound colonisation" OR "wound contamination") AND ("wound care" OR "wound care instructions" OR "wound hygiene" OR "surgical site care" OR "bandage change") AND ("post caesarean section mothers" OR "cesarean mothers " OR "post-operative mothers" OR "post caesarean birth" OR "cesarean birth").

2. The keywords for searching included the following MeSH terms ("caesarean section recovery" OR "attaining health" OR "post-partum recovery" OR "caesarean section wound healing) AND ("nutritional recommendation" OR "nutritional guide" OR "diet" OR "nutrition" OR "home care interventions") AND ("post caesarean section mothers" OR "cesarean mothers " OR "post-operative mothers" OR "post caesarean birth" OR "cesarean birth").

3. The keywords for searching included the following MeSH terms ("surgical site infection" OR "SSI" OR "wound sepsis" OR "wound infection" OR "wound colonisation" OR "wound contamination") AND ("hand hygiene" OR "bathing" OR "general hygiene" OR "perineal care" OR "hygienic recommendation") AND ("post caesarean section mothers" OR "post-operative mothers" OR "cesarean mothers " OR "post caesarean birth" OR "cesarean birth").

4. The keywords for searching included the following MeSH terms ("caesarean section recovery" OR "attaining health" OR "recovery" OR "postpartum recovery" OR "caesarean

wound healing") AND ("recommended exercise" OR "walking" OR "exercise" OR "Kegels exercise" OR "abdominal exercise") AND ("post caesarean section mothers" OR "cesarean mothers " OR "post-operative mothers" OR "post caesarean birth" OR "cesarean birth").

5. The keywords for searching included the following MeSH terms ("caesarean section recovery" OR "attaining health" OR "recovery" OR "postpartum recovery" OR "caesarean wound healing") AND ("home environment" OR "environmental ventilation" OR "environmental light" OR "environmental health" OR "environmental cleanness") AND ("post caesarean section mothers" OR "cesarean mothers " OR "post-operative mothers" OR "post caesarean birth" OR "cesarean birth"). Manual searching by scanning the list of references from the selected papers and searching for guidelines from the ministry of health was also done.

*Searching or sampling of literature*. The literature will be systematically searched from the following databases: PubMed, HINARI and Google scholar; from the inception of the databases to April 2021. This search will include empirical literature (research articles), and secondary literature (books, protocols and guidelines) that provide components, instructions or procedures of post-CS home care with a focus on preventing SSI and enhancing recovery. The search will include interventional studies, observational studies and qualitative studies. Two reviewers will independently screen the titles and abstracts using the inclusion and exclusion criteria and agree on the selected papers through discussion. The third reviewer will resolve any disagreements that rose between the two reviewers.

*Data collection*. A validated tool by Ursi and Gavao [32] will guide data extraction to ensure that the collection of relevant data from all the retrieved documents is performed, the risk of transcription errors is minimised, and the precision in data collection is guaranteed [31].

*Critical analysis*. All of the retrieved documents that meet the inclusion criteria will critically be analysed. The analysis of post-CS home care will be classified based on the type of study design used as suggested by Ursi (2005) and are as follows: Level 1: evidence from a meta-analysis of multiple randomised controlled clinical trials; Level 2: evidence from individual studies with experimental design; Level 3: evidence from quasi-experimental studies; Level 4: evidence of descriptive (non-experimental) studies or with a qualitative approach; Level 5: evidence from case reports or experience and Level 6: evidence based on opinions of specialists [32]. Data from different types of studies, guidelines and protocols will be collectively interpreted to understand the home care guideline and its effect on the rate of SSI.

*Discussion of the results and presentation*. The results will be discussed as well as the applicability of the recommendations to the post-CS mothers in Tanzania. Followed by their presentation at a scientific conference, they will be published in a relevant journal. The integrative review is intended to provide important components and content of home care for post-CS mothers that exist in the literature as a baseline for developing the home care guidelines.

## Phase II: Qualitative study

### Objectives.

1. To explore the experiences of post-CS mothers on the education they were provided by nurse midwives and the care they received at home

2. To describe the experiences of nurse-midwives on home care information provided to post-CS mothers.

3. To explore the experiences of caretakers while providing care to post-CS mothers at home after hospital discharge.

## Methodology

**Design.**    A descriptive exploratory study design [33] will be chosen to explore the experiences of nurse-midwives, post-CS mothers and their caretakers on post-CS home care.

**Setting.**    This qualitative study will be conducted in Dodoma at Makole Health Centre. Makole Health Centre serves a larger population with maternal and child health services (i.e. reproductive and child health clinic, antenatal and post-natal clinic) compared to other health facilities in Dodoma Region. In 2020, Makole health centre recorded a total of 7328 deliveries with a CS rate of 13.6%, and about 900 and 150 clients attended monthly antenatal and post-natal clinics, respectively (anecdotal evidence).

**Participants and sampling.**    This qualitative study will involve nurse-midwives who had worked in the post-natal wards for at least one year, assuming they had adequate experience with the normal routine of the ward including the provision of health education to post-CS mothers. Further, the study will involve mothers who recently had a CS delivery and attended a post-natal clinic for a check-up to explore their experiences after receiving education from nurse midwives and the care they received at home. Also, caretakers of post-CS mothers at home will be involved in the study. However, post-CS mothers who experienced prolonged hospitalisation due to maternal or foetal complications will be excluded. The purposive sampling technique [34] will be used to recruit between 5–25 participants [30], however, the principle of data saturation [35] will guide the sampling process.

**Data collection.**    Semi-structured, face-to-face, interviews with nurse midwives, post-CS mothers, and caretakers. Nurse midwives, post-CS mothers and caretakers will be explored their experiences with home care information provided to post-CS mothers, the education they were provided by nurse midwives, and the care they provided to post-CS mothers respectively. Interviews will be done using Swahili semi-structured interview guide [30]. Swahili is the national language spoken fluently by both researchers and participants. The interview will be carried out by the first author (MM) in the hospital environment, in a private quite room away from hearing reach of other people. Interviews will be recorded with permission from the participants ensuring that necessary information needed in the study is captured [30].

Before the interview is conducted, participants will be requested to provide written informed. The interviewer will start by interviewing nurse midwives who will also assist in identifying the post-CS mothers based on the inclusion criteria. After interviewing nurse midwives, the interview to Post-CS mother will follow and lastly the to caretakers. Caretakers will be obtained by visiting post-CS mothers at their homes, after making an appointment with them during the day of their interview at the post-natal clinic. There will be an audio recording, observation and written transitions (documentation) to ensure that all information is properly captured.

**Data analysis.**    Content data analysis s described by Graneheim and Landman [36] will guide the analysis. The audio recorded interviews will be listened to several times and then transcribed verbatim without translation to restore the natural meaning of participants' account. Initial coding will be done by first and second authors by using the first two interviews transcript independenlty. Meaning units will be extracted and condensed by shortening the original text, while maintaining the primary meaning. These condensed meaning unit will be condensed further to codes and grouped into sub categories, categories and themes. All authors will discuss and agree on the final codes and themes before the code book is developed. The code book will be used to analyse other transcript of the following interviews, new codes indentified from other transcript will be added to the code book after discussion and agreed by all authors. The reflect of the coding process will be done through out the analysis process [36]. The ATLAS.ti software will be used to organise data and to ensure themes are assigned

without distorting the meaning of the raw data. To establish confirmability, the results of the study will be returned to the participants for them to prove whether the study's findings represent the participants' responses. Thereafter, the themes of this study and findings from the integrative review will be incorporated to develop the draft guide (prototype 1) [37].

### Phase III: Developing post-CS home care guide

**Objective.** To explore expert opinion on the content of post-CS home care guide prepared through the review of literature, and the opinions of post-CS mothers and nurse midwives

*Design.* Participatory Action Research (PAR) will be applied to the development of a post-CS home care guide [38]. The findings from the integrative review (phase I) and qualitative study (phase II) will be used to prepare a draft of the post-CS home care guide (prototype 1). The draft guide will be reviewed by maternal and child health experts for appropriateness and adequacy of the post-CS home care guide.

A one-day workshop will be conducted to validate prototype 1. Participants in this workshop will be obstetricians and gynaecologists, nurse midwives, nutritionists, health promotion specialists and community health specialists. The participants will be obtained from their professional associations. Three representatives will be requested from each of the Association of Obstetrics and gynaecology (AGOTA), Tanzania Midwives Association (TAMA), Food and Nutrition Association of Tanzania (FONATA) and Public Health Association (TPHA). Further, the Ministry of Health, Community Development, Gender, Elderly and Children (MHCDGEC) will also be requested to provide three representatives making a total of 15 participants. During the workshop, the prototype 1 home care guide will be presented by the first author (MM) section by section and the process that led to the formation of the guide will be also shared. The aim is to present how the authors came up with the prototype I of the guide. Thereafter, three (3) focus group discussions will be conducted where each group will be led by a moderator and another person will take notes.

Each group will be composed of one member from each speciality. Participants in each group will discuss the draft guide and give recommendations on each component and content of the guide (prototype 1). The participants will discuss each section separately. They will start with the first section and give their opinion and recommendations before moving to the following section. They will do so using a discussion guide until the entire draft of the home care guide is discussed. They will be requested to give reasons for recommending or not recommending the items within the guide. Then they will present the recommendations in a plenary where consensus will be reached [39]. In the end, the participants will be asked for any additional comments or opinions on previous sub-sections to wrap up the session. Prototype 1 will then be revised based on the ground recommendations to form prototype 2.

### Pre-testing the prototype II post-CS home care guide and tool

The prototype II of the post-CS home care guide will be pretested with nurse-midwives and post-CS mothers in the post-natal ward by the researcher, to assess the feasibility and applicability of the guide in a real clinical setting. Also, the pre-test will involve the assessment of the tool that will be developed based on the designed home care guide; on its ability to measure knowledge of home care after hospital discharge for both nurse midwives and post-CS mothers. The pre-test will employ a pre-and post-interventional assessment for both nurses and midwives. The pre-test study will take place in the Morogoro region, in Morogoro regional referral hospital (MRRH).

A convenient sample of post-natal mothers will be obtained based on their availability during the data collection period and provided written consent to participate in the study.

Thereafter, mothers will be trained by a researcher using the developed post-caesarean home-based guidelines (prototype II). However, before the intervention, participants will be assessed for their baseline knowledge about home care after CS. During the intervention, various training strategies will be used including lecturing and lecture discussion. Following the intervention, post-natal mothers will be asked to return to the facility after 7 days after CS for post-training interventional assessment. In each visit to the post-natal clinic, mothers will be requested to provide feedback and suggestions on the adequacy of the education content, clarity of the information, and applicability of the recommendations in their homes, and to provide additional information that could be included in the care of mothers at home after caesarean section deliveries (if any).

The post-caesarean home-based guidelines (prototype II) will then be used to train nurse midwives after pre-assessment of knowledge on post-CS home care is conducted. The training intervention will involve 50 purposive recruited nurse midwives (i.e., 20% of the total sample size of nurse midwives in phase 4) [40] who are working in maternity wards at MRRH for at least one year; in the meeting room through lecturing method and slide projection. The aim is to familiarise the nurse midwives or midwives with the prototype II post-CS home care guide. Post-test to nurse midwives will be done on the same day after pretesting and intervention. The nurse midwives will then be required to teach post-CS mothers about home care before discharge as per the guideline to ensure its integration into the healthcare system and sustainability.

The pre-interventional knowledge assessment will be compared to the post-interventional knowledge assessment to evaluate change in knowledge before and after the intervention. Not only that but also nurse midwives will also be required to grade items of the guide by using the Likert scale (very important, important and less important); at the end of the pre-test study (phase 3), it is estimated that there will be recommendations on each item of the guide. They will also be required to give suggestions on the adequacy of the guide, clarity of the item, the applicability of the guide in hospital settings and proper timing of counselling post-CS mothers.

**Analysis.** In this study, data will be analysed by using the Statistical Product for Service Solutions (SPSS) software programme version 25.0. Before conducting the analysis, error checking (data cleaning) will be performed by using Frequency Distribution Tables to ensure the completeness and accuracy of data.

Descriptive analysis will be used to analyse participants' characteristics to determine the frequencies and percentages of their distribution. Descriptive statistical analysis will also be used to establish frequencies and percentages of the Likert scale responses among nurse midwives at the end of the pre-test study to determine the prevalence of their recommendations in each item. Knowledge of the participants will be categorised into two responses, adequate if the participants score above the mean of the knowledge question and inadequate knowledge if the participants score below the mean [41]. Logistic regression will be used to determine the association between the predictor variables such as age, education level, marital status, gravidity and parity and participants' knowledge about home care after CS set at a confidence interval of 95% with a margin of error of 5%. The baseline and end-line timepoints will be used to establish the effect of the intervention on knowledge.

The results from the pre-test study will help to improve the guide and to form the post-CS section home care guide (prototype 3), which will be used in phase IV of the study.

## Phase 4: Implementation and effectiveness assessment of post-CS home care guide

**Objectives.** 1. To assess the implementation process of post-CS home care guide in preventing SSI among mothers delivered by CS in health facilities in central Tanzania.

2. To evaluate the effectiveness of a post-CS home care guide in the prevention of SSI among post-CS mothers delivered at health facilities in central Tanzania.

3. To evaluate the effectiveness of a post-CS home care guide in improving knowledge of home care after hospital discharge among nurse midwives and post-CS mothers at health facilities in central Tanzania.

## Methodology

**Study design.**   Quasi-experimental study design with the non-equivalent control group (pre-and post-test with a control group) will be conducted [42]. The design will involve a baseline assessment of knowledge of post-CS and nurse midwives on home care after hospital discharge both in the intervention and control groups [42]. The intervention group will be exposed to a post-CS home care guide and education for both post-CS mothers and nurse midwives, respectively. In contrast, the nurse midwives and post-CS mothers in the control group will not be exposed to the post-CS home care guide and education but they will receive the usual care offered in their facility.

**Setting.**   This phase will be conducted in Dodoma and Singida. In Dodoma, the study will be conducted in Dodoma Regional Referral Hospital (DRRH), where an intervention group (nurse midwives and post-CS mothers) will be obtained. The DRRH provides both Basic Emergency Obstetrics and Newborn Care (BEmONC) and Comprehensive Emergency Obstetrics and Newborn Care (CEmONC) services. It has a separate maternity block for providing maternal and child health services. DRRH will be purposively selected as an interventional hospital because of the recently reported high rate of SSI due to CS [19]. In 2020, DRRH had 9016 total deliveries and the rate of CS was 28.7% (anecdotal evidence).

In Singida, the study will take place at Singida Regional Referral Hospital (SRRH). In SRRH, the control group will be obtained. The SRRH matches with DRRH in the sense that both are referral hospitals in the central zone of Tanzania where most of the complicated cases are managed. Both hospitals provide CEmONC including CS, and they have separate maternity blocks. In 2020, SRRH had a total of 3656 deliveries with a CS rate of 33.5% (anecdotal evidence).

**Study population.**   The study will include registered nurse midwives and post-CS mothers. In addition, post-caesarean mothers in post-natal wards will be retained to participate in the study before discharge. The exclusion criteria will be:

1. Post-CS mothers with prolonged hospitalisation due to maternal or neonatal complications.

2. Post-CS mothers who didn't receive pre-antibiotic medication prior to surgery, have an infection of amniotic fluid.

3. Mothers whose CS were not performed in the study site (DRRH and SRRH).

Post-CS mothers in the intervention group will be matched based on their age, gravidity, type of CS and their obstetric history such as the previous mode of delivery. Also, nurse midwives will be matched based on their education level and working experience in a maternity unit.

**Sample estimation.**   *Sample size for nurse midwives*. The sample size for the intervention and control group of nurse midwives will be obtained by using the formula for comparing two independent samples (intervention group vs control group), guided by the study by Joho [40]. In her study, which is conducted in 39 primary health facilities in Dodoma, she found that the knowledge proportion of nurses on the timing of administration of uterotonic before and after

the educational intervention was 44.8% and 63.4%, respectively [40,43].

$$n = 2\{Z_\alpha\sqrt{[\pi_0(1-\pi_0)]} + Z_\beta\sqrt{[\pi_1(1-\pi_1)]}\}/(\pi_{1-}\pi_0)^2$$

n = Minimum sample size
Zα = Standard normal deviation of 1.96 at 95% confidence interval (CI).
Zβ = Standard normal deviation 0.80 with the power demonstrating statistically significant difference before (baseline) and after intervention between two groups at 80%.
p0 = Proportion of an intervention group at baseline = 44.8% (0.448)
p1 = Proportion after the intervention = 63.4% (0.634)
With a 5% attrition rate:

$$n = 2\{1.96\sqrt{[0.634(1-0.634)]} + 0.8\sqrt{[0.448(1-0.448)]}\}/(0.634-0.448)^2)$$

n = 78
So, the minimum sample size will be 78 plus a 5% attrition rate = 82
The ratio of the intervention group to the control group will be 1:2. Therefore, the sample size for the control group will be 164.
Thus, the total sample size in this study will be 246 nurse midwives.
*Sample size for post-CS mothers*. The sample size for the intervention and control group of post-CS mothers will be obtained by using the following formula;
The formula for comparing two independent samples (intervention group vs. control group), and using the proportion of SSI for pre-intervention was 16% and 4.9% for post-intervention [44]. A prospective observational cohort study involving active in-patient and post-discharge surveillance cohort study was done in DRRH at Obstetrics and Gynaecology Department where [43];

$$n = 2\{Z_\alpha\sqrt{[\pi_0(1-\pi_0)]} + Z_\beta\sqrt{[\pi_1(1-\pi_1)]}\}/(\pi_1-\pi_0)^2$$

n = Minimum sample size
Zα = Standard normal deviation of 1.96 at 95% confidence interval (CI).
Zβ = Standard normal deviation 0.8 with the power demonstrating statistically significant difference before (baseline) and after intervention between two groups at 80%.
p0 = Proportion of an intervention group at baseline = 16% (0.16)
p1 = Proportion after the intervention = 4.9% (0.049)
With a 5% attrition rate:

$$n = 2\{1.96\sqrt{[0.16(1-0.16)]} + 0.8\sqrt{[0.049(1-0.049)]}\}/(0.16-0.049)^2$$

n = 145 plus 5% attrition rate 152
So, the minimum sample size is 145 plus a 5% attrition rate = 152
The ratio of the intervention group to the control group will be 1:2. Therefore, the control group sample size will be 304.
Therefore, the total sample size in this [40] study will be 456 post-CS mothers.

**Sampling and method.** A purposive sampling method will be used to select DRRH and SRRH [42]. SRRH and DRRH are both regional hospitals, have specific blocks for maternity care, and offer CEmONC. The intervention has been assigned by hospital settings where DRRH will be selected to be an intervention site because of its higher prevalence of SSI (48%) to post-CS mothers [19], and SRRH will serve as a control site.

Nurse-midwives and post-CS mothers will be selected by using a simple random sampling method [40]. Specifically, lottery replacement methods will be employed where the

replacement selection of participants will be done to ensure an equal chance of selection to all until the desired sample size is achieved [40].

**Intervention.** The intervention implementation of the post-CS home care guide will follow the CFIR framework, specifically on the implementation process by considering four factors, namely: planning, engaging, executing, and reflecting and evaluating [27]. In the planning stage, the researcher will meet with DRRH management to explain the purpose and schedule of the intervention. After the meeting with the management, the researcher will engage the nurse in charge of the wards on how to execute the training and mobilise nurse midwives and post-CS mothers.

*Assessment and education of post-caesarean mothers.* Before the intervention, the pre-tested tool will be used by the researcher and RA to assess baseline knowledge of post-CS mothers on home care after CS. Both the assessment and intervention of post-CS mothers will be done before discharge. The education intervention about home care will be given in one group depending on the number of post-CS discharged on that day. The intervention will involve slides projection of the post-CS home care and distribution of leaflets with simplified Swahili language and pictures demonstrating recommended home practices after CS. This will be followed by the questions and answers session for more clarification. The intervention will take about 1 hour including a question-and-answer session. This will be performed every day before discharge until the sample size is met. The researcher and other RA (nurse midwives in the profession) will be providing this education to post-CS mothers to ensure the home care guide after CS is delivered effectively. Thereafter, the participants will be instructed to come for the usual recommended post-natal visits to the same facilities for follow-ups that is, on the 7[th], 14[th] and 30[th] day. On the 7[th] day post-delivery, mothers will be instructed to come for stitch removal, assessment of the wound and other routine post-natal check-up services. In all visits, the post-CS mothers will be requested to report their practices at home, and assess their haemoglobin level to establish their anaemia status using the WHO classification [45]. For the purpose of this study, mothers will be instructed to come down to the facility on the 30[th] day post-CS to evaluate their prognosis, to assess wound healing and their recovery in general. The wound assessments will be performed by two researchers and research assistants to minimise observer bias. For all wounds with SSI, their samples will be taken for culture to confirm the diagnosis. On the same day (30[th]-day post-delivery), a post-interventional assessment of the knowledge of post-CS home care will be done [46], after being exposed to the home environment.

*Assessment and training of nurse midwives.* The intervention by nurse-midwives will be followed after the intervention of the post-CS woman. They will also be trained in a written post-caesarean home care guide and its use the guide to educate post-natal mothers. Before the training, the pre-tested tool will be used to assess the baseline knowledge of nurses on post-CS home care. Thereafter, nurse midwives will be trained on the component and content of the guide and how they would use the guide to educate post-CS mothers. The intervention will be done in the hospital environment (meeting room) to ensure maximum participation of the study participants. The session will involve lecturing and lecturing discussion. The intervention will involve two training sessions scheduled on two different days since it is difficult to get all of the nurse-midwives in one day. The intervention will take three hours including the time for pre-interventional assessment. Thereafter, they will be post-tested on the same day. After the intervention, nurse midwives working in the post-natal ward will be required to provide home care education intervention to post-CS mothers, for making this intervention sustainable.

At SRRH, the control group of nurse midwives and post-CS mothers will not be exposed to the content of the developed home care guide. Nurse midwives will continue to provide their

usual care to post-CS mothers at the facility. The control group will also be assessed for baseline knowledge of home care after CS. Post-CS mothers will be assessed first, they will be assessed for baseline knowledge about home care after hospital discharge before discharge, as in the interventional group.

Knowledge of participants on home care after CS and SSI status will be compared between the intervention and control group. To ensure equal treatment of both groups, nurse midwives in the control group will be trained on the post-CS home care guide and how to deliver the content to post-CS mothers after meeting the sample size of post-CS, where they will also be required to train post-CS mothers to ensure the sustainability of the intervention. The implementation of the intervention has been described in the Fig 3.

**Variables definition and measurement. Independent variables:** In this study, the independent variables are socio-demographic characteristics of participants such as age, marital status, education level, past obstetric history, type of operation and economic factors for the post-CS mothers as well as the as age, sex, education level and duration of experience for nurse midwives.

## Intermediate independent variables

**Hb level.** Hb level will be measured using HemoCue Hb 201+ Haemoglobin photometer, which uses a capillary to collect blood after a figure prick. Participants will be classified as non-anaemic ($\geq$ 11g/dl), mild (9.5–10.9g/dl), moderate (8–9.4g/dl), severe (6.5–7.9g/dl) or life-threatening ($<$ 6.5g/dl) anaemia according to standard WHO criteria [45].

**Reported home care practice.** This care regimen is received by mothers at home after hospital discharge as per the post-CS home care guide. Mothers who report performing/receiving 75% of home care practice as per the guide will be regarded as having good practices, otherwise poor practice [47].

**Dependent variables. Knowledge of post-CS home care.** Participants who will score at least 75% of the maximum score will be regarded as possessing adequate knowledge of home care after CS, otherwise inadequate knowledge [47].

**SSI.** This involves inflammation and positive bacterial culture from the wound within 30 days after CS [48]. It can be associated with redness, pain around the wound area, foul odour, and any discharge from the wound [49, 50]. The diagnosis and classification of SSI wound infection will be based on CDC criteria within 30 days [48] at the interval of 7th, 14th and 30th days after CS during post-natal clinic visits. A Blueberry Wound Healing Questionnaire (WHQ) will be used to assess signs and symptoms of SSI [49]. The tool comprises signs and symptoms for a 'not at all' (score 0), 'a little' (1), 'quite a bit' (2) and 'a lot' (3), where the higher score will suggest an SSI. Further, in order to minimise observation bias, two research assistants will assess the wounds. The confirmation is made through microbiological culture and isolation of microbes which aid in the management of the wound.

**Analysis.** In this study, data will be analysed by using the Statistical Product for Service Solutions (SPSS) software program version 25.0. Before conducting the analysis, data will be cleaned using frequency distribution tables to ensure the accuracy of data entry. Continuous data will be treated for normality to determine the measure of analysis (parametric and non-parametric analysis).

## Analysis of demographic data

Descriptive analysis will establish frequencies and percentages of socio-demographic characteristics profiles of the study participants and will be presented in tables and figures in order to establish distribution proportions between groups.

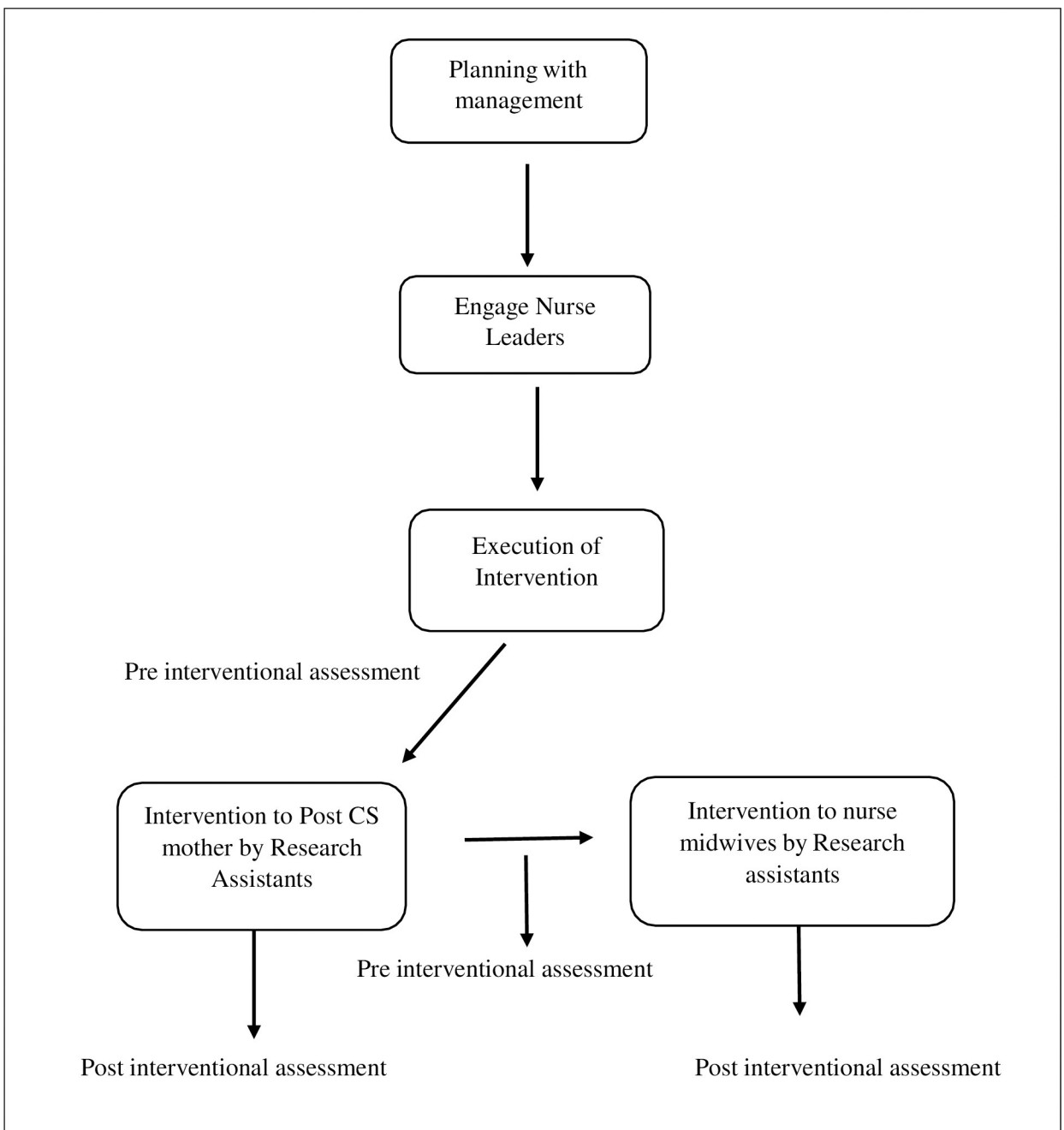

**Fig 3. Description of implementation of intervention.** Involvement of the hospital administration in interevention phase before enrolling the post CS mothers and nurse midwives to the study.

The Chi-square test will be used to analyse the categorical data and to establish the relationship between the categorical variables and the outcome variables that are post-intervention knowledge on home care after CS and SSI.

## Analysis of knowledge of home care after CS

The outcome variable is dichotomised (1 = adequate knowledge; 0 = inadequate knowledge), and so a generalised estimating equation (GEE) will be employed to assess the impact of the

intervention on knowledge, whereby odds ratio, confidence interval and p-value will be presented. GEE is an extension of the ordinary logistic regression model for correlated/repeated measure data. In this case, it seems to be more appropriate as will be having more than one measurement/observation of each subject at different time points.

## Analysis of SSI

Binary logistic regression analysis will be used to assess factors associated with SSI.

The general logistic regression model is given as:

$$\log it[\pi(x)] = \log\left(\frac{\pi(x)}{1 - \pi(x)}\right) = \beta_0 + \beta_1 x_1 + \ldots\ldots + \beta_p x_p$$

Where $\pi(x)$ is the SSI status, $x_i's$ is set of independent variables and $\beta_i's$ is their respective parameters.

The power of the study will be set at 80% (0.8) while a confidence interval of 95% with a margin of error of 5% (0.05) was used as a statistical measure of significance ($< 0.05$ was regarded as significant while $> 0.05$ not significant).

The effect of intervention between groups will be assessed using difference in difference (DID) analysis.

**Ethical issues.** Prior to the start of the study, ethical clearance and approval will be obtained from the Muhimbili University of Health and Allied Sciences (MUHAS) Institutional Board Review (IRB). Then, permission to conduct the study will be obtained from TAMI-SEMI, Regional Medical Office (RMO), District Medical Office (DMO) and lastly, from the medical officer and ward in charge, where the study participants will be obtained. The informed consent will be obtained and signed by all participants after the aim of the study has been explained to them, along with benefits and harm (if any). The confidentiality of the participants in the data will be highly maintained. To achieve this, confidential identification numbers will replace the names of post-CS mothers and will only be available to the PI and supervisors. Mothers who will have SSI and post-partum anaemia will be referred to be reviewed by a doctor for further treatment and follow-up.

Nurse midwives and post-CS mothers in the control group will be exposed to the guide after the study. This will be done to ensure equal benefits for both interventional and control groups.

## Discussion

The establishment of this guide will also help to cement the need for pre-discharge health education counselling for nurse midives to post-CS mothers and the entire community on home care practices. This will help Ministry of Health to incorporate the guide national wide, with the aim of improving home care after CS na reducing CS complications such as SSI. The limitations of the study include the risk of desirability bias, as participants may not feel comfortable giving honest feedback about their care. This will be prevented by ensuring post CS mothers that their responces will not be shared with their nurse mideives or caretakers hence they should feel free to share them with researcher. Another limitation is the the risk of SSI, which might be increased by other factors like multiple vaginal examinations during labour, time of rupture of membrane, duration of operation, existing medical condition of a woman like HIV/AIDs and diabetes, anaemia status, lack of antibiotics prophylaxis during the pre-and post-operative period and infection in the amniotic fluid. To mitigate this, adequate information regarding labour and operation procedure will be considered from the patient's file and medical history is recorded from post-CS mothers, which will be used in the inclusion criteria

of study participants and data analysis. have been adjusted by the statistical analysis in the statistical model.

The study findings will be published in reputable journals and presented at scientific conferences. The guide produced in this study will be shared with the Tanzania Ministry of Health, for possible adoption on the training of post-CS mothers in health facilities. The final dissertation report will be submitted to MUHAS and UDOM libraries.

## Acknowledgments

On behalf of all of the authors, I appreciate the University of Dodoma for funding my PhD study.

## Author Contributions

**Conceptualization:** Mwajuma Bakari Mdoe, Lilian Teddy Mselle, Stephen Mathew Kibusi.

**Methodology:** Mwajuma Bakari Mdoe, Lilian Teddy Mselle, Stephen Mathew Kibusi.

**Supervision:** Lilian Teddy Mselle, Stephen Mathew Kibusi.

**Writing – original draft:** Mwajuma Bakari Mdoe.

**Writing – review & editing:** Mwajuma Bakari Mdoe, Lilian Teddy Mselle, Stephen Mathew Kibusi.

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
