## [Decision Letter · Decision Letter 0]

3 Aug 2022

PONE-D-21-40464

Designing and testing the effectiveness of a post caesarean section home care guide in preventing postpartum complications in Central Tanzania: Protocol

PLOS ONE

Dear Dr. Mdoe,

Thank you for submitting your manuscript to PLOS ONE. After careful consideration, we feel that it has merit but does not fully meet PLOS ONE’s publication criteria as it currently stands. Therefore, we invite you to submit a revised version of the manuscript that addresses the points raised during the review process.

We look forward to receiving your revised manuscript.

Kind regards,

Jennifer Yourkavitch

Academic Editor

PLOS ONE

Journal Requirements:

3. Thank you for stating the following in the Acknowledgments Section of your manuscript: "On behalf of all authors we acknowledge the redness of University of Dodoma for funding the study."

Please remove any funding-related text from the manuscript and let us know how you would like to update your Funding Statement. Currently, your Funding Statement reads as follows: "The study is funded internally by the University of Dodoma, scholarship to the first author (MBM). However, the funder had and will not have a role in study design, data collection and analysis, decision to publish, or preparation of the manuscript"

Additional Editor Comments (if provided):

Thank you for this protocol for a study investigating a very important issue. Please respond to all points raised by the reviewers.

The Academic Editor was one of the reviewers.

Please also have someone edit this to ensure correct English grammar. Standard proofreading (fixing small errors related to tense and number) would eliminate many errors.

Reviewers' comments:

Reviewer's Responses to Questions

**Comments to the Author**

1. Does the manuscript provide a valid rationale for the proposed study, with clearly identified and justified research questions?

Reviewer #1: Yes

Reviewer #2: Yes

2. Is the protocol technically sound and planned in a manner that will lead to a meaningful outcome and allow testing the stated hypotheses?

Reviewer #1: Partly

Reviewer #2: Partly

3. Is the methodology feasible and described in sufficient detail to allow the work to be replicable?

Reviewer #1: Yes

Reviewer #2: No

4. Have the authors described where all data underlying the findings will be made available when the study is complete?

Reviewer #1: No

Reviewer #2: Yes

5. Is the manuscript presented in an intelligible fashion and written in standard English?

Reviewer #1: No

Reviewer #2: No

6. Review Comments to the Author

You may also provide optional suggestions and comments to authors that they might find helpful in planning their study.

Reviewer #1: Thank you for the opportunity to review the paper titled “Designing and testing the effectiveness of a post caesarean section home care guide in preventing postpartum complications in Central Tanzania; Protocol.” I am very enthusiastic about the proposed research as I agree that there is a dearth of research on this topic and currently poor practice in this area.

Unfortunately, there are numerous errors in writing throughout the manuscript. Below, I highlight some of these issues on the first few pages, but as the authors will note, there are so many that this was not possible to do for the entire paper. I strongly recommend engaging in a peer-editor to clean this up so that the very valuable content is not lost in minor errors in writing.

Even with concerns in the writing, the content is clear enough for my enthusiasm to remain. I do highlight a few technical points for them to consider in this next stage.

1. Reference 2 is dated.

2. More facility delivery is the only reason there is more c/section (last line first paragraph, page 4), and so that needs to be expanded. Then when I looked into the reference to understand what that paper is saying, reference 3 is for “Five national academies call for global compact on air pollution…” This is clearly not the right reference.

3. To that end – please check *ALL* for your references. Many of them are incorrectly formatted.

4. In addition, I want to call your attention to a few new articles recently out that should be included. None of these negate this proposal. If anything, it supports the claim that this area is critically neglected.

- Discharge instructions given to women following delivery by cesarean section in Sub-Saharan Africa: A scoping review. Musabeyezu et al. https://doi.org/10.1371/journal.pgph.0000318

- New global WHO postnatal guidance is welcome but misses the long-term perspective. Bick et al.

- Also look up the WHO guidelines referenced in the Bick et al paper.

5. The search criteria need a lot of work. Some examples:

- If I just copy and paste the first bullet into pubmed, nothing comes up. That is concerning.

- There are many ways to spell cesarean; your terms only look at one.

- By looking at “post cesarean section” together in quotes, then you miss just cesarean. I write a lot about post-c-section surgical site infections. In none do I write “post cesarean section” exactly.

6. You need more detail than “hand searching”

7. Will you use multiple coders in your qualitative analysis?

8. Be careful – it is not a rickets scale.

9. Can you provide the tool described on Page 16 for knowledge assessments?

10. For your sample size calculation, Your z_beta is for 80% power; for 90% power, you need 1.282. Fine to just clarify and power at 80% power.

11. For your sample size calculation – not totally unreasonable, but not conventional to put at exact observed levels from other studies. It is fine to keep, but also fine to let those studies guide you but not completely drive what you do.

12. I really don’t expect that your intervention will cut the SSI rate by 2/3. (From 16% to 4.9%). Personally, I would not power for this as: 1) Even if a good intervention, this is not the sole (or even primary) driver of SSI. 2) Even if you don’t have significant declines in SSI rates, good messaging/communication to moms has lots of benefits.

13. You power for binary measures, but then your primary analysis (page 24) treats scores as continuous. Either need to change your sample size calculation approach or your analysis approach.

14. Editing issues found on the first few pages. (As noted above, it was not possible to make this level of editing throughout, but this is hopefully illustrative of what will need to be done throughout.)

• Page 2, second line. Should be plural “infections”. Also needs to be SSIs. Need to check throughout the paper, as anywhere there is SSI it may, or may not, need to be written as SSIs.

• Page 2, make validation plural: “Following a series of validations…”

• Page 4, first line, need a comma between intervention and performed. “…. Intervention, performed to…”

• Page 4, change from into to in, to read “… may result in complications…”

• Page 4, “…hospitalization, increase helath cost,…” add ‘d’ to increase, fix spelling, add s to cost, to read “… hospitalization, increased health costs,…”

• Page 4, add “and” to “… for her baby, and maternal death.”

Reviewer #2: Please edit for English grammar.

This study has a nice design--the mixed methods approach is appropriate.

Please clarify "intervention" on p. 8: "...evaluation of the intervention in increasing knowledge of

home care and preventing postpartum complications." Are you evaluating the training or the guide? How would you separate training and guide in this circumstance.

p.11: I don't understand this sentence: "Data extraction will be guided by the validated to ensure collection of relevant data from all retrieved document, risk of errors in transcription is minimized, to guarantee precision in data

collection and to serve as a record (30)."

p. 12 "Purposive sampling technique (32) will be used to recruit 5-25 participant (Creswell, 2016) based on principle of data saturation (33)." Which participants? midwives or mothers? Do you intend to conduct interviews until you've reached saturation?

You could consider making full transcriptions of interviews for accuracy and easier coding.

For the post-test with mothers, will you also evaluate if the recommended practices were applied?

I don't think 30 mothers are enough to use multivariable regression. Your question for multivariable analysis is not clear. What is the exposure? All mothers were counselled on the new guidance. (Phase 3).

On p. 17 you introduce prevention of post-partum anemia. But the purpose of the guide to that point is to prevent SSI. Preventing, identifying, treating anemia involves a different protocol. Many factors contribute to anemia.

Why not randomize mothers to the intervention?

How will you choose controls (the non-equivalent control group)? Will you match on certain characteristics? That decision has implications for the analysis.

How will you prevent "contamination" in the education and training of midwives to those not involved with the intervention?

The mothers cared for by the intervention midwives will be the intervention mothers. For controls, you will have to enroll 2 mothers cared for by non-intervention midwives, correct? Text on p.23 indicates the intervention assignment is by hospital; I do not see that explained earlier in the protocol.

To mitigate other causes of SSI, you might create a scoring system based on info in patients' files that indicates risk of SSI from issues other than home care. You could then control for that score.

The dependent variables you list in the figure--shouldn't they be just knowledge and SSI? Are you following mothers to determine if SSI develops? For how long will you follow them?

State the dependent and exposure variables clearly for each quantitative assessment.

7. PLOS authors have the option to publish the peer review history of their article (what does this mean?). If published, this will include your full peer review and any attached files.

Reviewer #1: No

Reviewer #2: No

---

## [Author Response · Author response to Decision Letter 0]

16 Sep 2022

Responses to comments made by Reviewers:

Authors thank the reviewers for the constructive comments that in our view we believe will improve our work to meet the quality standards of this prestigious journal. Regarding the issue raised by editor

1. The editorial issues has been solved by the work being sent to English editor for clarity

2. The caption and reporting of supporting files has been corrected as per guideline

3. References has been updated accordingly

Further responses have been written in the author response documents

4. More details about the methodology especially on analysis have added for more clarity.

Note: Kindly receive the responses of the authors to the raised comments of reviewers, we would like to clarify more in case of unsatisfactory response; aiming at improving our work.

Thank you very much

---

## [Decision Letter · Decision Letter 1]

25 Oct 2022

PONE-D-21-40464R1Designing and testing the effectiveness of a post caesarean section home care guide in preventing surgical site infection in Central Tanzania: ProtocolPLOS ONE

Dear Dr. Mdoe,

Thank you for submitting your manuscript to PLOS ONE. After careful consideration, we feel that it has merit but does not fully meet PLOS ONE’s publication criteria as it currently stands. Therefore, we invite you to submit a revised version of the manuscript that addresses the points raised during the review process.

We look forward to receiving your revised manuscript.

Kind regards,

Jennifer Yourkavitch

Academic Editor

PLOS ONE

Additional Editor Comments (if provided):

Please respond to all reviewer comments. Please number the lines in all versions of the manuscript.

Note that the Academic Editor is Reviewer #2 and those comments are included here.

These suggestions refer to the tracked changes version:

1. P. 17-18: Say more about the purposive sampling of midwives (what criteria) and the convenience sampling of post CS women (between certain days or something else?).

2. You said that you removed references to anemia but it is still in the Phase 4 Objective 1 and there is reference to hemoglobin level and anemia status on pages 26 and 27. Please recheck the entire document to remove all references to anemia.

3. I see the addition of matching criteria for post-CS women. What are the matching criteria for nurse midwives?

4. Phase 4 dependent variable--should it be SSI incidence? Is knowledge measured by a score? How will SSI be measured? By count? An outcome variable that is a count should be analyzed with Poisson regression. The knowledge score could be analyzed with linear regression. It's not clear why you are using logistic regression.

5. If you are assessing the outcomes at three time points, then you need to account for time in the statistical models. You could use a time-to-event analysis for SSI, for example. You could consider a time course analysis or a generalized estimating equation. Please ask a statistician to review your methods. The approach to analyzing SSI should be described separately from the approach to analyzing the change in knowledge.

Reviewers' comments:

Reviewer's Responses to Questions

**Comments to the Author**

1. Does the manuscript provide a valid rationale for the proposed study, with clearly identified and justified research questions?

Reviewer #1: Yes

2. Is the protocol technically sound and planned in a manner that will lead to a meaningful outcome and allow testing the stated hypotheses?

Reviewer #1: Partly

3. Is the methodology feasible and described in sufficient detail to allow the work to be replicable?

Reviewer #1: No

4. Have the authors described where all data underlying the findings will be made available when the study is complete?

Reviewer #1: Yes

5. Is the manuscript presented in an intelligible fashion and written in standard English?

Reviewer #1: No

6. Review Comments to the Author

You may also provide optional suggestions and comments to authors that they might find helpful in planning their study.

Reviewer #1: Overall, I have seen improvements in the revised document, but note that the paper still has considerable typos and grammar issues to be resolved. These are largely in newer sections, so perhaps that wasn’t reviewed?

1. There is a disconnect for me in this statement: “In Tanzania mainland, the prevalence of CS is 6% (2); although anecdotal evidence shows the facility rate of CS to be as high as 48%. This can be explained by increased proportion of deliveries occurring at the health facilities (3).” I cannot figure out how increased facility deliveries would result in facility c-section rates of 48%.

2. Remove the , from “… guideline further, admits paucity of studies…”. (That entire paragraph has lots of typos and needs close editing)

3. Note, in this sentence: “Studies have shown that no or less information is provided to post CS mothers regarding home care after CS (13)”. This is more accurately “A study in Haiti showed that…” Also, when I looked up reference, and I saw that reference 13 has Harvard misspelled.

4. On page 7, there is a hanging “To design”. Not sure what that is for?

5. You have maintained keeping the “post” in all of your search terms despite my previous comment. When you search that first line, you get “Your search was processed without automatic term mapping because it retrieved zero results.”. These terms *will not work*.

6. The authors reference a validated tool, but don’t provide it: “Data extraction will be guided by the validated tool…” (I believe the reference provided is for an approach, not a tool.) Are they going to develop and validate a tool? If so, this needs to be clarified.

7. As you discuss developing the guide, note this new consensus guidelines has since been published and should be referenced. Kateera et al, “Safe recovery after cesarean in rural Africa: Technical consensus guidelines for post-discharge care”. These are meant to be a guide still needing further adaptation, and so the steps you outline still apply. But it would be a failing to not start with this initial work.

8. For the Phase 3 analysis, I am very confused. The authors reference binary outcomes (logistic regression, Chi-squared tests) but I cannot tell what is the binary outcome. They also reference a continuous outcome, but this is more obvious – changes in scores.

9. You have left anemia in Objective 1 of Phase 4, but have responded to the reviewer that you have removed anemia from the study.

10. I previously commented on my disbelief that this intervention could drop SSIs by 2/3, and the authors responded that this was guided by effect sizes observed in another study. But that study had a **totally** different intervention and study population. I maintain my earlier comment – this intervention is important, but don’t overstate its ability to impact SSIs given the complexity of where SSI risk sits.

11. You need to put the “Analysis of categorical data” as a header

12. The second reviewer suggested creating SSI risk scoring system to control for in analysis. You responded to their comment, but I didn’t see these changes reflected in the document. I am not entirely sure this is a good idea, because particularly for east Africa, I am not sure these risks have been well studied.

7. PLOS authors have the option to publish the peer review history of their article (what does this mean?). If published, this will include your full peer review and any attached files.

Reviewer #1: No

---

## [Author Response · Author response to Decision Letter 1]

12 Dec 2022

Thank you for your comments aimed at improving our work. Implementation of intervention has been amended in the current documents that; instead of nurse midwives to educate the post CS mother, post-CS mother will be assessed and intervened and by research assistants. This is because post CS mother being cared with the intervened nurse midwives will increase the risk of being taught/told home care by intervened nurse midwives in interventional than in control group; leading to unequal comparable groups between control and intervention hospitals. Before we planned to do pre-interventional knowledge assessment 48 hours after CS, the risk exposing post CS mother to the intervention was thought to be high difficult to control hence research assistants will assess and train post-CS mothers, followed by nurse midwives

---

## [Editor Report · Decision Letter 2]

12 Jan 2023

PONE-D-21-40464R2Designing and testing the effectiveness of a post caesarean section home care guide in preventing Surgical Site Infection in Central Tanzania; ProtocolPLOS ONE

Dear Dr. Mdoe,

Thank you for submitting your manuscript to PLOS ONE. After careful consideration, we feel that it has merit but does not fully meet PLOS ONE’s publication criteria as it currently stands. Therefore, we invite you to submit a revised version of the manuscript that addresses the points raised during the review process. Please submit your revised manuscript by Feb 26 2023 11:59PM. If you will need more time than this to complete your revisions, please reply to this message or contact the journal office at plosone@plos.org. Please include the following items when submitting your revised manuscript:A rebuttal letter that responds to each point raised by the academic editor and reviewer(s). You should upload this letter as a separate file labeled 'Response to Reviewers'.A marked-up copy of your manuscript that highlights changes made to the original version. You should upload this as a separate file labeled 'Revised Manuscript with Track Changes'.An unmarked version of your revised paper without tracked changes. You should upload this as a separate file labeled 'Manuscript'.If applicable, we recommend that you deposit your laboratory protocols in protocols.io to enhance the reproducibility of your results. Protocols.io assigns your protocol its own identifier (DOI) so that it can be cited independently in the future. For instructions see: https://journals.plos.org/plosone/s/submission-guidelines#loc-laboratory-protocols. Additionally, PLOS ONE offers an option for publishing peer-reviewed Lab Protocol articles, which describe protocols hosted on protocols.io. Read more information on sharing protocols at https://plos.org/protocols?utm_medium=editorial-email&utm_source=authorletters&utm_campaign=protocols.

We look forward to receiving your revised manuscript.

Kind regards,

Jennifer Yourkavitch

Academic Editor

PLOS ONE

Journal Requirements:

Additional Editor Comments (if provided):

Please upload the response to reviewers for Revision #2, which you emailed to the academic editor.

You use the terms "mother", "woman", and "female" throughout the manuscript. Please use one term consistently. I suggest using "woman."

Please again review for English grammar word usage and punctuation. It gets better with each revision but there are still approximately two errors on each page. For example, there are several instances where you use a semi-colon incorrectly, as in the Abstract methodology section. The journal does not provide editing services so you want your submitted manuscript to be correct.

There are 5 descriptions of search terms but only four questions (p.10). It's not clear why you would conduct 5 separate searches.

P. 11--"data extraction will be guided by the validated tool..." What is the tool you are referring to here?

P. 29 study power. You say 80% but then have (0.84). Is that a typo?

Do you want to add a Limitations section to the Discussion? If so, please include the possibility of uncontrolled confounding in the estimates derived from the statistical models.

Reviewers' comments:

None

---

## [Author Response · Author response to Decision Letter 2]

1 Mar 2023

On behalf of all other, we are thankful for receiving the constructive comments from the reviewers. 

Below is the table of comments and their responses.

S/N COMMENTS RESPONSE PAGE

1. Please review your reference list to ensure that it is complete and correct. If you have cited papers that have been retracted, please include the rationale for doing so in the manuscript text, or remove these references and replace them with relevant current references. Any changes to the reference list should be mentioned in the rebuttal letter that accompanies your revised manuscript. If you need to cite a retracted article, indicate the article’s retracted status in the References list and also include a citation and full reference for the retraction notice. 1. The references have been edited; two references have been removed because are retracted references (number 10 and 30). 

2. Most have been improved and the link or ‘doi’ number have been added, in Mendeley if you delete the in text citation the reference is automatically deleted hence it is not seen as strikethrough

3. 31-37

2. You use the terms "mother", "woman", and "female" throughout the manuscript. Please use one term consistently. I suggest using "woman."

 The term woman has been used throughout the manuscript Entire document

3. Please again review for English grammar word usage and punctuation. It gets better with each revision but there are still approximately two errors on each page.

For example, there are several instances where you use a semi-colon incorrectly, as in the Abstract methodology section. The journal does not provide editing services so you want your submitted manuscript to be correct. The English grammar, word usage and punctuation have been revised in the entire document Entire Document

4. There are 5 descriptions of search terms but only four questions (p.10). It's not clear why you would conduct 5 separate searches. The fifth question was missing; it has been added.

Regarding five separate search; this is because if you search home care after caesarean delivery you will find not results until you separate the care is when you will find articles. I tried that, after failing I got an advices to search items separately 9

5. P. 11--"data extraction will be guided by the validated tool..." What is the tool you are referring to here? The tool has been adopted from Elizabeth Ursi and Cristina Gavao as cited by Marcela Souza. The citation has been added in the documents and attachment of the tool has been added in the submission site 11

6. P. 29 study power. You say 80% but then have (0.84). Is that a typo? Yes it is a typo, and it has been corrected from 0.84 to 0.80 21,22,28

7. Do you want to add a Limitations section to the Discussion? If so, please include the possibility of uncontrolled confounding in the estimates derived from the statistical models. The limitation section have been added 30

The Validated tool used to collect the integrative review data

Data collection instrument (By Ursi, 2005).

A. Identification 

Title of the article 

Title of the journal 

Authors 

Country 

Language 

Year of publication 

B. Institution holding the study 

Hospital 

University 

Research center 

sole institution 

Multicenter study 

Other institutions 

No identification of setting 

C. Type of publication 

Nursing publication 

Medical publication 

Publication in another area of health. Which area? 

D. Methodological characteristics of the study 

1. Type of publication 

2. Objective or investigation question 

3. Sample 

4. Treatment of Data 

5. Intervention performed 

6. Results 

7. Analysis 

8. Implication 

9. Evidence level 

E. Assessing methodological rigor

Clarity to identify the methodological steps in the text (method employed, participating subject, inclusion/ exclusion criteria, intervention, results) 

Identification of restriction of Bias

---

## [Editor Report · Decision Letter 3]

12 Mar 2023

PONE-D-21-40464R3Designing and testing the effectiveness of a post caesarean section home care guide in preventing Surgical Site Infection in Central Tanzania; ProtocolPLOS ONE

Dear Dr. Mdoe,

Thank you for submitting your manuscript to PLOS ONE. After careful consideration, we feel that it has merit but does not fully meet PLOS ONE’s publication criteria as it currently stands. Therefore, we invite you to submit a revised version of the manuscript that addresses the points raised during the review process.

We look forward to receiving your revised manuscript.

Kind regards,

Jennifer Yourkavitch

Academic Editor

PLOS ONE

Journal Requirements:

Additional Editor Comments (if provided):

Please see Word file with suggested edits and a few questions.
---

## [Author Response · Author response to Decision Letter 3]

11 May 2023

Editorial Comment

1. Please ensure that you refer to Figure 1 & 2 in your text as, if accepted, production will need this reference to link the reader to the figure

Response: Thank you for the comments, citation of S1_Fig has been removed as instructed previously. We have replaced them by referring to fig 1, 2 and 3 (page 6,8 and 26)

2. Please upload a new copy of Figure 1 & 2 as the detail is not clear. Please follow the link for more information: https://blogs.plos.org/plos/2019/06/looking-good-tips-for-creating-your-plos-figures-graphics/

Response: The figure has been uploaded and the new version has been submitted in the system

3. Please include a separate legend for each figure in your manuscript.

Response: Separate legends for each figure have been added to the manuscript. Figure captions have also been added in the manuscript as per authors instructions of the journal (page 6,8,26,35-37).

Response: - The source of fund or this study is the University of Dodoma, that covered both the tuition fee for the PhD and the data collection costs.

- Mdoe Mwajuma B. and Kibusi, Stephen M. are the employees of the University of Dodoma and they receive their salary from this institution.

---

## [Editor Report · Decision Letter 4]

29 May 2023

Protocol for designing and testing the effectiveness of a post caesarean section home care guide in preventing surgical site infection in Central Tanzania

PONE-D-21-40464R4

Dear Dr. Mdoe,

We’re pleased to inform you that your manuscript has been judged scientifically suitable for publication and will be formally accepted for publication once it meets all outstanding technical requirements.

Kind regards,

Jennifer Yourkavitch

Academic Editor

PLOS ONE
---

## [Editor Report · Acceptance letter]

5 Jun 2023

PONE-D-21-40464R4 

Protocol for designing and testing the effectiveness of a post caesarean section home care guide in preventing surgical site infection in Central Tanzania 

Dear Dr. Mdoe:

I'm pleased to inform you that your manuscript has been deemed suitable for publication in PLOS ONE. Congratulations! Your manuscript is now with our production department. 

Kind regards, 

on behalf of

Dr. Jennifer Yourkavitch 

Academic Editor

PLOS ONE